# Role of Gut Microbiota in Hepatocarcinogenesis

**DOI:** 10.3390/microorganisms7050121

**Published:** 2019-05-05

**Authors:** Haripriya Gupta, Gi Soo Youn, Min Jea Shin, Ki Tae Suk

**Affiliations:** Division of Gastroenterology and Hepatology, Institute for Liver and Digestive Diseases, Hallym University College of Medicine, Chuncheon 24253, Korea; phr.haripriya13@gmail.com (H.G.); gisu0428@gmail.com (G.S.Y.); wehome3@hallym.ac.kr (M.J.S.)

**Keywords:** hepatocellular carcinoma, gut microbiota, gut–liver axis, intestinal dysbiosis

## Abstract

Hepatocellular carcinoma (HCC), one of the leading causes of death worldwide, has a causal nexus with liver injury, inflammation, and regeneration that accumulates over decades. Observations from recent studies have accounted for the involvement of the gut–liver axis in the pathophysiological mechanism responsible for HCC. The human intestine nurtures a diversified colony of microorganisms residing in the host ecosystem. The intestinal barrier is critical for conserving the normal physiology of the gut microbiome. Therefore, a rupture of this barrier or dysbiosis can cause the intestinal microbiome to serve as the main source of portal-vein endotoxins, such as lipopolysaccharide, in the progression of hepatic diseases. Indeed, increased bacterial translocation is a key sign of HCC. Considering the limited number of clinical studies on HCC with respect to the microbiome, we focus on clinical as well as animal studies involving the gut microbiota, with the current understandings of the mechanism by which the intestinal dysbiosis promotes hepatocarcinogenesis. Future research might offer mechanistic insights into the specific phyla targeting the leaky gut, as well as microbial dysbiosis, and their metabolites, which represent key pathways that drive HCC-promoting microbiome-mediated liver inflammation and fibrosis, thereby restoring the gut barrier function.

## 1. Introduction

Liver cirrhosis and hepatocellular carcinoma (HCC) constitute the most chronic forms of liver disease and are designated as end-stage liver diseases. With a mortality of 9.1% worldwide, HCC is the fifth most common cancer and is considered a significant global health burden [1]. Chronic viral hepatitis, especially hepatitis B virus (HBV) and hepatitis C virus (HCV), is the leading cause of the pathophysiological progression of HCC [2]. However, other etiologies, such as drug abuse, autoimmunity, intake of liver toxins, alcohol, and nonalcoholic fatty liver disease (NAFLD), are also correlated with a high risk of HCC [3].

Recently, a dramatic relationship was observed between the microbiome and HCC using a newly developed diagnostic method [4]. The gut milieu is comprised of numerous bacteria in addition to archaea, eukarya, and viruses, all of which play essential roles in maintaining the homeostasis and vital functions of a healthy host by generating active metabolites. These microbe-derived metabolites connect the gut microbiome to the circulatory, immune, and hormone systems through signaling to the host metabolism [5,6]. Intestinal bacterial growth promotes diseases in confined local areas as in the case of chronic inflammatory bowel disease, as well as in remote areas which include the liver, heart, brain, skin, and hematopoietic systems [7]. The liver is very closely associated with the gut given its anatomical location. Since the liver receives the majority of its blood and nutritional supply from the gut through the portal vein, it is the first organ to be exposed to gut-derived toxic factors, including bacteria, damage-associated metabolites, i.e., damage-associated molecular patterns (DAMPs), and bacterial products (i.e., pathogen-associated molecular patterns (PAMPs) [8,9]. Alteration of the intestinal microbiome leads to disruption of the intestinal wall and promotes the increased translocation of bacteria and their active metabolites, PAMPs, an event that often cause systemic inflammation, known as endotoxemia. This, however, has a significant effect on the progression of chronic hepatic injuries, which include NAFLD and alcoholic liver disease (ALD) [6,10]. These complications are often preterminal events in cirrhosis, and their prevention and early management could improve the prognosis of affected patients and the further progression to HCC. 

The gut microbiota has emerged as a paramount causative event in the progression of hepatic malignancies. Henceforth, being the most pliable organ in human, gut microbiota should be targeted to procure eubiosis. Integrity of microbiome is a desideratum by the therapeutic gut modulation which can be retrieve by therapeutic interventions. Therefore, this review provides an update on the various mechanisms that may show acrimonious communication between the liver and gut microbiota, and how their modulation during pathogenesis contributes to the progression of hepatic diseases to HCC. 

Research searches were performed in five global databases (Cochrane, Embase, PubMed, PsycINFO, and NDSL) up to January 2019, with search terms focused on the population with HCC and the terminology of microbiota.

## 2. The Gut Microbiota: A Diverse Colony of the Microbiome and Its Microbial Dysbiosis

The human gastrointestinal tract is inhabited by a distinct array of bacteria and other microorganisms that have a symbiotic relationship with the host, known as the gut microbiota [11]. The human gut microbiota is monopolized by the bacterial phyla Bacteroidetes and Firmicutes, which comprise 90% of the total microbiota. These are accompanied by Actinobacteria, Proteobacteria, Verrucomicrobia, Fusobacteria, and Cyanobacteria [12,13]. The microbial density represents 1.5–2 kg of biomass, which is dominated by anaerobic bacteria that increase in density near the distal edge of the intestine [14]. The autochthonous bacteria create a broad range of metabolites that function as important signaling and energy substrates for cells that cover bile acids, such as deoxycholic acid and lithocholic acid, and short-chain fatty acids, such as acetate and butyrate. Since butyrate is a key substrate in cell metabolism, it is the prime energy molecule for colonocytes [15]. These metabolites play distinct roles in nourishing the colonic mucosal cells, suppressing local colonic inflammation, and maintaining glucose homeostasis and energy regulation, thus affecting colon physiology [9,16,17]. The composition of the intestinal barrier is important in maintaining the physical separation of function between the microbiome and the host. The microbiota influences immune mediated intestinal barrier function and hence regulates the access of metabolites to the portal circulation and the liver [15]. 

Under normal conditions, there is an optional passage of metabolites through the intestinal epithelium; however, obstruction to the intestinal barrier provokes increased bacterial translocation and increased leakage of bacterial metabolites [18]. The microbiota can also have an impact on the elimination of bacterial pathogens from the liver by the activation of Kupffer cells or by tolerance induced by portal vein antigens [19].

## 3. Pathophysiological Factors in Hepatocarcinogenesis

HCC is a highly complex and heterogeneous disease that affects all populations across the globe. The incidence of HCC may vary due to regional and geographic differences in the pervasiveness of causal factors [20]. HCC has been linked to a multitude of etiological risk factors and cofactors; in approximately 80–90% of patients, cirrhosis precedes HCC [21,22]. Of the myriad factors associated with HCC, most eminent factors include HBV and HCV infection, chronic alcohol consumption, and DNA change [2]. 

Another factor that emerged in the past decade is gut dysbiosis. Irrespective of their prominence, disrupted gut barrier function suggests consequences for hepatic cell damage. Moreover, some evidence has shown a link between altered gut microbiota and increased intestinal permeability that can lead to disease progression at various stages, and might promote the progress of HCC throughout all these stages [7]. 

Below, a brief discussion focuses on the most prevailing risk factors for HCC and, undoubtedly, the common underlying causes of cirrhosis that have been determined as crucial risk factors of HCC [23,24]. However, HCC can occur in non-cirrhotic livers, which accounts for approximately 20% of all HCC cases [25]. Figure 1 illustrates the comprehensive factors, studied extensively, that direct the progression of HCC. 

### 3.1. Virus, Microbiota, and Hepatocellular Carcinoma

Hepatitis-related viruses, such as HBV and HCV, are strongly correlated with the development of HCC [20]. Hepatic virus-induced hepatocarcinogenesis is generally a multistep process, which may include cellular inflammation, the induction of oxidative stress, and interference with signaling pathways, causing the targeted activation of oncogenic pathways [26] and genome integration of the virus into host DNA via host DNA deletion [27]. These viruses continually reproduce in culture and show non-cytopathic behavior, despite the fact that HCV may also show cytopathic behavior [28,29,30]. Furthermore, without the complete elimination of viruses from the host, relentless replication induces inflammation, which perpetuates chronic liver disease and thus poses a risk factor for HCC [31]. 

Concerning mechanistic insights into the viruses responsible for the dysbiosis of microbiome-mediated HCC, there is limited information. However, interrelated studies have suggested that the gut microbiota upholds the pathophysiology of viral hepatitis and may progress to advanced stages of HCC. The establishment of the gut microbiota greatly affects the immune response of the liver, leading to either the elimination or persistence of the virus. 

Treatment with antibiotics prevents the clearance of HBV in adult mice and it has been indicated that the immune-tolerating pathway dominates through Toll-like receptor (TLR)-4-dependent innate immunity. Meanwhile, the absence of TLR-4 might impede the progress of liver tolerance, which was observed in TLR-4-deficient mice that did not manifest tolerance and rapidly cleared HBV [32]. Moreover, a set of *Bifidobacterium* species was found to mark predominant dysbiosis in HBV cirrhosis patients [33]. Notably, in a clinical study, the HBV level in patients was positively correlated with disease progression and the risk of developing HCC [34]. TLR-4 induction and activation are considered to mediate carcinogenesis by a synergistic effect of alcohol and HCV nonstructural protein 5A. A progenitor stem cell marker and TLR-4 downstream gene were also upregulated upon the activation of TLR-4 receptor and aided TLR4-dependent liver carcinogenesis [35]. Microbiome remodeling was also seen in HCV patients, which was conceivably altered by bacterial translocation [36]. 

Hence, the gut microbiota might control antiviral responses that are involved in disease progression and HCC development. Taking this into account, clinical studies have provided apparent data showing that HBV appeared to have increased in LPS in HCC patients [37] and that it altered fecal microbial content in cirrhosis patients [38]. 

### 3.2. Alcohol, Microbiota, and Hepatocellular Carcinoma

ALD comprises asymptomatic steatosis, steatohepatitis, advanced and accelerated fibrosis, and cirrhosis, and super-positioned HCC covers a wide range of diseases. Up to 90% of patients with excessive alcohol consumption usually have reversible asymptomatic steatosis upon abstinence [39,40]. However, persistent alcohol consumption can cause inflammation in the liver, termed alcoholic hepatitis. Eventually, hepatic fibrosis deposition (20%–40%) and liver cirrhosis (8%–20%) can develop with a high risk of HCC [41,42,43]. 

The mechanisms underlying ALD pathogenesis include the production of reactive oxygen species directly induced by the liver, ethanol, and its metabolites; the activation of innate immunity (lipopolysaccharide (LPS)–TLR4 signaling, and the complement system); and the production of inflammatory cytokines such as tumor necrosis factor (TNF)-α [44,45]. Chronic alcohol consumption increases intestinal permeability, leading to high levels of endotoxins, such as LPS [46], which is produced by Gram-negative bacteria. LPS is transported directly through the hepatic portal vein, which acts as a pivotal mediator of inflammation in ALD. It also enables the production of reactive oxygen species and TNF-α activation by Kupffer cells and leads to inflammation or injury to the liver. In addition, these pro-inflammatory cytokines and LPS cause the release of excess amounts of collagen and α-smooth muscle actin, which activate hepatic stellate cells and further promote fibrosis [47,48,49,50]. 

The important contribution of the gut microbiota to early stages of ALD has been established in previous studies. It is evident that increased levels of plasma LPS are associated with different stages of ALD-fatty liver, hepatitis, and cirrhosis, which is further explained by increased intestinal permeability [51]. Animal studies have demonstrated that alcohol feeding disturbs the intestinal environment, thereby reducing the synthesis of long-chain fatty acids [52]. TLR^−/−^ and gut sterilization with antibiotics lead to reduced hepatic steatosis and inflammation [53,54], signifying that the interplay between gut microbiota and TLR-4 is important for promoting ALD.

The functional processes of the gut microbiota–TLR-4 axis in advanced liver diseases, i.e., cirrhosis and HCC, are not well understood, possibly due to complications and obstacles involved in the animal model of ALD. Additionally, tumor development was inhibited in ethanol-fed TLR-4^−/−^ mice, which further proved that the sustained activation of TLR-4 in alcohol-fed mice induces HCC in synergy with HCV [35]. These studies are consistent with established clinical observations in patients with chronic HCV infection, whereby excessive intake of alcohol is an important cofactor that leads to the development of advanced liver diseases and HCC [7,55]. A systemic review collated clinical data which demonstrated that alcoholic cirrhosis patients have worsened dysbiosis and different relative abundances of microbiota. Lachnospiraceae and Ruminococcaceae were found to be less abundant in cirrhosis patients, while Enterobacteriaceae was found at a relatively high level in such patient. Enterobacteriaceae and Streptococcaceae were found to be associated with alcoholic hepatitis [56]. A clinical study on hospitalized Alcoholic hepatitis (AH) patients showed that a plentitude of *Streptococcus*, *Bifidobacterium*, *Enterobacter* and *Atopobium* species are correlated with severe alcoholic hepatitis [57]. 

Fecal microbial transplantation in mice from the studied patients showed distinct bacterial genera compositions. *Bilophila*, *Alistipes*, *Butyricimonas* and *Clostridium* cluster XIVa were more abundant in mice with severe alcoholic hepatitis, whereas *Barnesiella*, *Parasutterella* and an unclassified Alphaproteobacteria genus were more abundant in nonalcoholic hepatitis mice. Also, *Akkermansia muciniphila*, *Howardella*, *Phascolarctobacterium*, *Faecalibacterium prausnitzii*, *Turicibacter*, *Desulfovibrio*, and *Gemmiger* were found almost exclusively in nonalcoholic hepatitis mice microbiota, though they exhibited low abundancy in severe alcoholic hepatitis mice [57]. These findings were further corroborated by different studies reporting that patients with severe alcoholic hepatitis had elevated levels of Actinobacteria and reduced levels of Bacteroidetes [58]. Other clinical studies also demonstrated the lower abundance of *Akkermansia muciniphila* in alcoholic hepatitis, while the oral administration of *Akkermansia muciniphila* in animal models was found to ameliorate the integrity of the intestinal barrier [59]. 

### 3.3. Nonalcoholic Fatty Liver Disease, Microbiota, and Hepatocellular Carcinoma

NAFLD amounts to an array of pathological conditions, from fatty liver to nonalcoholic steatohepatitis (NASH). During the process, steatosis is likely to be mild; however, hepatocyte injury (ballooning), inflammation, and peri-cellular fibrosis are distinctive features of NASH, which is likely to lead to cirrhosis, liver failure, and HCC. In addition, patients with NAFLD are at increased risk of developing HCC, even in the absence of liver cirrhosis [60]. The underlying pathophysiology of NAFLD and in particular multifactorial NASH and is strongly linked to insulin resistance, aberrant hepatic lipid metabolism, visceral adiposity, and inflammation. Recently, studies have shown that the intestinal microbiota also plays a significant role in the pathogenesis of NAFLD [9,61]. In addition, obesity induces changes in the composition of the gut microbiota and its metabolites (LPS or PAMPs) [62]. DAMPs from dying hepatocytes induce the movement of inflammatory molecules by TLR and inflammasome activation in target immune cells and stimulate the transition from NAFLD to NASH [63]. NASH-specific fibrosis pathways are driven by the activation of hepatic stellate cells, which is the main event in hepatic fibrosis; these cells are also responsive to stimulation by various metabolites that are present in a diseased fatty liver [64,65]. 

Although NAFLD is associated with a relatively low individual risk of HCC development, due to its predominance, it significantly contributes to HCC development at the population level [66]. Numerous studies have shown that the intestinal microbiome is sensitive to the intestinal wall and modulates homeostasis. Changes in the integrity of the intestinal tract can be observed by the disruption of tight junctions and the increased permeability of NAFLD biopsy patients [67]. Dysbiosis has been observed in patients with NAFLD; however, studies have demonstrated differences in patterns with the microbial environment [68,69]. The microbial environment is significantly involved in the progression of NAFL to NASH, which was not induced by long-term loading of exogenous LPS in mice [70]. Dysbiosis in mice fed a high fat diet resulted in low-level phosphatidylcholine, which impaired very low-density lipoprotein secretion, affecting the export of hepatic lipid, promoting fatty liver, and contributing to the development of NAFLD via choline metabolism shift [71]. Additionally, germ-free mice exhibited less HCC as compared to mice that had been subjected to chronic treatment with low doses of LPS, leading to a significant increase in HCC [72]. This suggests that antibiotic administration and intestinal sterilization can reduce both the initiation and progression of HCC in obese mice [73]. 

The potential impact of probiotics on decreasing hepatocyte injury is another factor in the inhibition of HCC in NAFLD [74]. Coadministration of a live multi-strain probiotic mixture with omega-3 fatty acids once daily for 8 weeks to patients with NAFLD reduce liver fat, improve serum lipids, metabolic profile, and reduce chronic systemic inflammatory state [75]. Combination of probiotics *Lactobacillus bulgaricus* and *Streptococcus thermophilus* attenuates liver aminotransferases in NAFLD patients [74]. Lowering of the proinflammatory marker could restrict the intestinal permeability and translocation of LPS in liver thus alleviating the development of NAFLD and NASH. In another report, the probiotic (Symbiter) reduces liver fat, aminotransferase activity, and the TNF-α and IL-6 levels in NAFLD patients [76]. Probiotics effectively prevent postoperative infections and improves early biochemical parameters of allograft function after liver transplantation [77].

The phylum Actinobacteria with the genera *Gemmiger*, *Parabacteroides* and *Paraprevotella* were abundant in early HCC as compared to liver cirrhosis. Moreover, in comparison with the control, the phylum Verrucomicrobia and the genera *Alistipes*, *Phascolarctobacterium* and *Ruminococcus* were decreased substantially while *Klebsiella* and *Haemophilus* were increased in early HCC in patients involved in a clinical study [78].

### 3.4. Genetic/Epigenetic Alterations, Microbiota, and Hepatocellular Carcinoma

In hepatic malignancies, metabolic and oxidative injury causes periodic inflammation, necrosis, and repetitive compensatory regeneration, and a high-throughput of hepatocytes promotes a progressive and steady accumulation of genetic errors, mutations, and epigenetic defects in cancer-related genes [79]. A close interaction between genetic and epigenetic alterations has been observed during cancer initiation and progression and is associated with the development of HCC [80]. Genetic alterations are irreparable modifications that can be observed early in precancerous stages of the cirrhotic liver. Early genetic mutations presumably initiate the developmental stage of HCC [81]. Genetic changes can be classified into many types, such as high chromosomal instability, and chromosome alterations, including telomere shortening, translocation, inversion, deletion, copy number variations, and nucleotide variations. At all levels, genetic changes typically lead to the loss of function of tumor suppressor genes that regulate activation or important oncogenes that regulate cell proliferation and growth [80,81]. Epigenetics are described as modifications of gene expression without altering the genetic code or the DNA sequence itself. Epigenetics regulate gene expression at the transcriptional or posttranscriptional levels. Alterations at the epigenetic level, such as DNA hyper-methylation or hypo-methylation, histone modification, chromatin remodeling, and aberrant expression of micro-RNAs and long noncoding RNAs, disrupt functional gene expression, which induces the abrupt activation of oncogenes or restrains the function of tumor suppressor genes, driving hepatocarcinogenesis [82]. 

In contrast to other liver diseases, there is little information on the role of the gut microbiota in epigenetic changes, which features an indirect relationship between the microbiome and epigenetic regulation through metabolites in the progression of HCC [83,84]. In a study, the gut microbiota influenced the regulation of epigenetics by immune homeostasis [85]. Dysbiosis of the microbial environment can interrupt homeostasis, thus triggering immune-mediated hepatocyte injury which further triggers HCC progression. This study illustrates that the immune response is pro-carcinogenic regardless of an insufficiency of cofactors, such as genotoxic agents or viral transactivation [86]. Histone deacetylases that modify chromatin structure regulate the transcription of genes and are involved in the process of carcinogenesis since histone deacetylates perform functions such as chromatin remodeling, gene repression, and cell cycle regulation. A high expression of histone deacetylase (HDAC1) was shown to be linked to aggression and cell dedifferentiation in HCC patients [87,88]. 

Short-chain fatty acids (SCFAs) inhibit the activity of HDAC [89], and a reduction in SCFAs can lead to the development of HCC. Ruminococcaceae (cluster IV), *Eubacterium* (cluster XIVa), and *Faecalibacterium* are dominant bacteria that produce SCFAs. However, decreases in SCFAs are associated with chronic liver diseases which then may progress to HCC [89,90,91]. Another important animal study deduced that Aflatoxin B_1_ (AFB_1_)-induced molecular alterations such as DNA damage or genotoxicity and oncogene expression in liver cells during carcinogenesis can be lowered by probiotic fermented milk (*Lactobacillus rhamnosus* GG and *Lactobacillus casei*) alone or in combination with chlorophyllin by reducing free radicals or superoxide anions generated by AFB_1_ [92].

## 4. Gut Microbiota and Hepatocellular Carcinoma

The role of the microbiota in hepatocarcinogenesis is mostly driven by inflammatory pathways, which are initiated by crosstalk between the intestinal bacteria, immune system, and liver. The process essentially involves the interplay of macrophages, Kupffer cells, and PAMPs in the liver. In the cascade of eliminating microorganisms, most populations of macrophages and Kupffer cells reciprocate to very low concentrations of PAMPs, endotoxins, or LPS via the activation of NF-κB by binding to TLRs, especially TLR-4 and -9, and nucleotide-binding oligomerization domain-like receptor (NOD-like receptor). This consequently leads to an inflammatory chain reaction that promotes inflammation and cytokine release [93]. Hence, dysbiosis of the gut microbiota boosts the secretion of inflammatory cytokines, such as TNF-α, IL-8, and IL1β, by Kupffer cells. IL-1β stimulates lipid accumulation and cell death in hepatocytes, causing steatosis and inflammation. Therefore, cytokines play a major role in the induction and progression of NAFLD to NASH and cirrhosis [94,95,96]. 

By altering bile acid metabolism, dysbiosis can promote the development of HCC in relation to NAFLD. A change in the composition of the gut microbiota is likely to lead to a high content of deoxycholic acid, which innervates the senescence-associated secretory phenotype of the hepatic stellate cells, resulting in the secretion of various inflammatory and tumor promoting factors, thus exacerbating the progression of HCC [73]. In a model of NASH-associated HCC induced by STHD-01 given to specific pathogen free (SPF) C57BL/6J mice, the accumulation of cholesterol and secondary bile acids caused hepatic inflammation and injury, which might contribute to enhanced carcinogenesis [97]. Additionally, Dapito et al. suggested that TLR-4 and the intestinal microbiota were not required for HCC initiation but for HCC promotion by mediating increased proliferation, the expression of the hepatomitogen epiregulin, and the prevention of apoptosis [72]. Gut sterilization confined to late stages of hepatocarcinogenesis reduced HCC, suggesting that the intestinal microbiota and TLR-4 represent therapeutic targets for HCC prevention in advanced liver disease [72]. Other animal studies demonstrated the key involvement of the microbiome in NASH aggravation. In addition, co-housing was found to further exacerbate NASH risk, though this was reduced by antibiotic treatments [98]. Sustained LPS accumulation was found to represent a pathological mediator of inflammation-associated HCC [99]. Probiotic treatment, prohep, slowed down tumor growth significantly and reduced tumor size by decreasing the Th17 cell level and the production of IL-17 in a mouse model of HCC [100]. Moreover, continuous administration of probiotics prior to liver transplant successively scaled down to 4.8% of 30-day post-transplant infection rate compared to the placebo in post-transplantation in patients [77]. 

In carcinogenesis, cytokines and T cells are important. The intestinal flora is critically involved in the pathogenesis of HCC by creating an anti-inflammatory microenvironment, which is dependent on liver LPS. *Alistipes*, *Butyricimonas*, *Mucispirillum*, *Oscillibacter*, *Parabacteroides*, *Paraprevotella*, and *Prevotella* were classified as enriched genera in this study, among which *Oscillibacter* species stimulate the differentiation of anti-inflammatory Treg cells that produce IL-10 and *Parabacteroides* species have proven to withhold inflammation by restraining inflammatory cytokines secretion and promoting the release of anti-inflammatory cytokines IL-10 [101,102]. Along with the aforementioned genera, species *Akkermansia muciniphila*, *Bacteroides fragilis*, *Parabacteroides distasonis*, and *Alistipes shahii* were also significantly enriched. *Alistipes shahii* tends to modulate the gut by abating tumor growth and *Bacteroides fragilis* acts by stimulating Treg cells for IL-10 production [100,103,104]. 

SCFAs derived from fermented dietary fibers also have a potential role in influencing cancer cell proliferation outside the gut; they increase the portal propionate level so as to prevent cancer cell proliferation in liver tissue [105]. A validated animal study established that pectin alleviates NAFLD by an intriguing mechanism of SCFAs [106]. Contrary to this, dietary fiber, viz. soluble and insoluble diets enriched with soluble fiber but not insoluble fiber, induced icteric HCC in dysbiotic mice. The inhibition of gut fermentation and the exclusion of dietary soluble fiber prevented HCC. Pharmacologic inhibition of the fermentation or depletion of fermenting bacteria markedly reduced intestinal SCFAs and prevented HCC. The class Clostridia, particularly *Clostridium* cluster XIVa and the phylum *Proteobacteria*, was determined to be firmly linked with HCC in this study [107]. A better prospective of fundamental processes such as dysbiosis, inflammation, and fermentation hay help in forming a strategy for preventing and treating conditions which lead to HCC (Table 1).

In clinical trials, the profile of the gut microbiota associated with the presence of HCC in cirrhotic patients is characterized by increased fecal counts of *E. coli*. Therefore, intestinal overgrowth of *E. coli* may contribute to the process of hepatocarcinogenesis [109]. Recently, non-HBV/HCV-HCC patients were found to harbor more potential pro-inflammatory bacteria (*Escherichia-Shigella, Enterococcus*) and reduced levels of *Faecalibacterium, Ruminococcus,* and *Ruminoclostridium*, resulting in a decrease in the potential of anti-inflammatory short-chain fatty acids [110]. In a previous report, the phylum Actinobacteria was increased in early HCC versus cirrhosis. Correspondingly, 13 genera including *Gemmiger* and *Parabacteroides* were enriched in early HCC versus cirrhosis [78]. *Bacteroides* and Ruminococcaceae were increased in the HCC group, while *Bifidobacterium* was reduced. *Akkermansia* and *Bifidobacterium* were inversely correlated with calprotectin concentration, which in turn was associated with humoral and cellular inflammatory markers [61] (Table 2).

## 5. Future Prospective for the Prevention of Hepatocellular Carcinoma

Understanding the etiology of bacterial pathogens that affect liver disease has led to attempts to manipulate microorganisms. Microbiota treatment could incorporate the utilization of probiotic, prebiotic, and synbiotic enhancements, or antimicrobials [111,112,113,114]. Antibiotics play an innate role in the treatment and prevention of cirrhosis complications. However, they can lead to problems by generating resistance. The most effective way to rehabilitate the gut microbiota is through diet, the incorporation of prebiotics and probiotics, or a combination of these strategies. These probiotics, prebiotics, and synbiotics produce intestinal benefits that influence host immunity, thereby restoring eubiosis and maintaining the integrity of the intestinal barrier by impeding the translocation of endotoxins. Additionally, remedial control of the tumor-related microbiome might also be acquired by fecal microbiota transplantation; interest has been growing for potential therapy, although promising results have yet to be reached. Changes in the physiology of bile acids that improve the function of intestinal barriers and favorably modulate the gut–liver axis are also areas for future therapeutic development. Future investigations ought to center around the metabolic capacity of the microbiota using metatranscriptomic and metabolomic approaches. In this manner, we can distinguish new metabolites produced by bacteria that provide more descriptive evidence of a bacterial role in liver disease.

## 6. Conclusions

HCC, which is a serious complication of cirrhosis, has shown contradictory evidence in terms of its relationship with the gut microbiota. Of the myriad components of the gut microbial habitat, inflammation is an important element that molds microbial composition. Intriguingly, whether microbial dysbiosis is perpetuated by the inflammatory cascade or by other factors that influence early microbial imbalance, which then propagate inflammation, is not yet evident. 

Current data from animal and clinical studies point in the direction of the gut-liver axis, showing promising results for the primary or secondary prevention of HCC. The microbiome provides a biomarker that can be tested for the risk of disease and its progression; nevertheless, it remains unknown whether it is the cause or outcome of the disease or whether it is an inferential risk factor or modulator of the disease. Therefore, these biomarkers hold promise for diagnostic and prognostic mechanisms that remain difficult to achieve. In light of the metagenomic revolution, research on the composition and function of the microbiome is an important goal to understand the development of cirrhosis as well as its progression to HCC. 

## Figures and Tables

**Figure 1 microorganisms-07-00121-f001:**
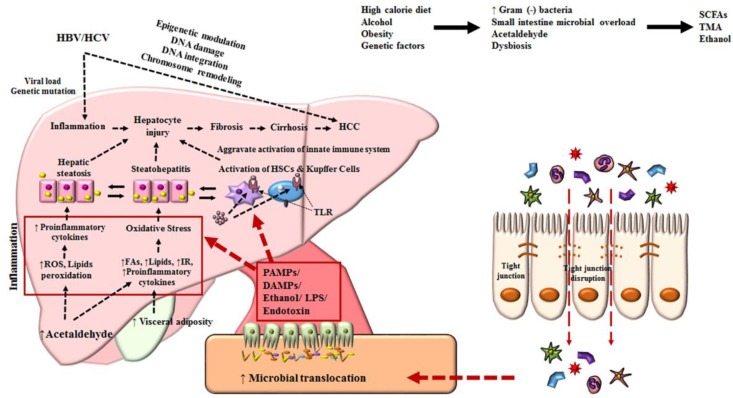
Mechanisms associated with the pathophysiology of hepatocellular carcinoma. Diet, alcohol, obesity, and genetic factors lead to prominent changes in microbiota which induce intestinal bacterial overgrowth, dysbiosis, intestinal permeability, bacterial translocation, and endotoxemia, resulting in the development of HCC. HCC, hepatocellular carcinoma; HBV, hepatitis B virus; HCV, hepatitis C virus; HSC, hepatic stellate cell; ROS, reactive oxygen species; TLR, Toll-like receptor; IR, insulin resistance; LPS, lipopolysaccharide; TNF, tumor necrosis factor; SCFA, short-chain fatty acid; DAMPs, damage-associated molecular patterns; PAMPs, pathogen-associated molecular patterns; TMA, trimethylamine.

**Table 1 microorganisms-07-00121-t001:** Animal studies of the relation between the microbiome and hepatocellular carcinoma.

Animal	Disease	Condition	Comparison	Biomarker	Microbiome Factor	Reference
C57BL/6 (p21-p-luc mice, age 4 weeks)	Obesity-HCC	Single DMBA at neonatal age followed by HFD for 30 weeks	Mice fed a normal diet vs. mice fed a high-fat diet	IL-6↑, p16↑, Gro-a↑, Ki-67↓, BrdU↓, cH2AX↑, CXCL9↑, 53BP1↑, Il-1b↑	*Clostridium* cluster XI and XIVa↑ *baiJ* gene↑	[73]
SPF C57BL/6J (age 8 weeks)	Nonalcoholic steatohepatitis (NASH)-HCC	STHD-01 diet 1 week after depletion of gut microbiota by a cocktail of Abx for 9 weeks or 41 weeks	STHD-01 fed mice vs. healthy mice	T3↑, ALT, AST↑, TNF-α, IL-1, Chol↑, Cyp 7a1↑	*Bacteroides*↑, *Clostridium* cluster XVIII↑, *Streptococcus*↓, *Bifidobacterium*↓, *Prevotella*↓	[97]
C57BL/6N (age 5–6 weeks)	HCC	Prohep (*L. rhamnosus* GG >5 × 10^9^, viable *E. coli* Nissle 1917 2.5–25 × 10^9^, and heat-inactivated VSL#3 (1:1:1)) for 38 days; tumor formation by Hepa 1–6, 1 × 10^7^ CFU	Prohep-fed mice vs. control group	Th17↓, FLT-1↓, ANG2↓, KDR↓, VEGFA↓, TEK↓, TGF-β↓, IL-17↓, RORγt↓, IL-27↑, IL-13↑, HIF-1↑	*Prevotella*↑, *Oscillibacter*↑, Treg/Tr1↑	[100]
Sprague–Dawley rats	HCC	Penicillin G sodium salt/DSS (0.3g/L) for 7 days for enteric dysbactriosis; DEN (40 mg/kg) once a week i.p. for 10 weeks; probiotics VSL#3, low (6 × 10^9^ CFU) and high (6 × 10^10^ CFU) doses given daily by gavage for 14 weeks	Probiotics + DEN vs. control group	ALT↓, HMGB1↓, Ki-67↓, NF-κB↓, IL-6↓, IL-10↑	LPS↓	[37]
C3H/HeOuJ, C3H/HeJ and C57Bl6	HCC	DEN (100 mg/kg) followed by biweekly injections of carbon tetrachloride (0.5 mL/kg i.p.); gut sterilization	TLR-deficient mice vs. wild-type group	Ki67↓, Pcna↓, Col1a1↓, Acta2↓, IL-6↓, TNF-α↓, CCL2↓, HGF↓, Epiregulin↓	LPS↓	[72]
Sprague–Dawley rats and C57BL/6 mice (age 6–8 weeks)	HCC	DEN (70 mg/kg weight) i.p. for 10 weeks; antibiotics, polymyxin B and neomycin, were added to drinking water 4 days prior to DEN injection until 3 weeks followed by 1 week of regular water until 10 weeks	Antibiotics + DEN vs. DEN group	IL-6↓, TNF-α↓, Ki67↓	LPS↓	[99]
Sprague–Dawley rats and C57BL/6 mice (age 6–8 weeks)	HCC	DEN (70 mg/kg weight) i.p. for 10 weeks; lethally irradiated; 1 × 10^7^ bone marrow cells injected i.v.; DEN treatment 5 weeks after transplantation	BMT in TLR4^−/−^ vs. BMT in wild-type mice	Ki67↓, phospho-c-Jun↓, Cyclin D1↓, ALT↓, IL-6↓, TNF-α↓, NF-κB↓		[99]
BALB/c mice (age 5 weeks)		Mice transplanted with Bcr-Abl-transfected BaF3 cells, received ITF in their drinking water	BaF3 vs. BaF3 + ITF	Malignant cell proliferation in liver tissue↓	*Lactobacillus* spp.↓	[105]
BCO1^−/−^BCO2^−/−^ double KO mice (male)	HCC	DEN (25 mg/kg b.w.) at 2 weeks old, followed by HFD from week 6 for 24 weeks; treatment: tomato powder (TP) for 24 weeks	DEN + HFD vs. DEN + HFD + TP	MCP1↓, iNOS↓, TNFα↓, IL1b↓, IL6↓, and IL12α↓, SIRT1, NAMPT	*Bacteroides*↓, *Mucispirillum*↓, *Clostridium*↓, *Parabacteroides*↓, *Lactobacillus*, *Bifidobacterium*↑	[108]

↑ indicates an increase in the condition of diseased/probiotics-treated group A relative to the condition of alcoholic disease B, ↓ indicates a decrease in condition A relative to condition B. *L. rhamnosus* GG, *Lactobacillus rhamnosus* GG; *E. coli*, *Escherichia coli*; DMBA, 7,12-dimethylbenz(a)anthracene; HFD, high fat diet; CFU, colony-forming unit; HCC, hepatocellular carcinoma; ALT, alanine transaminase; AST, aspartate aminotransferase; BAL, blood alcohol level; ALP, alkaline phosphatase; LPS, lipopolysaccharide; SREBP, sterol regulatory element-binding protein; TNF, tumor necrosis factor; IL, interleukin; CyP2E1, cytochrome P450 family 7 subfamily A member 1; VEGFA, vascular endothelial growth factor A; HIF, hypoxia-inducible factor; Bcl-2, B-cell lymphoma 2; IFNγ, interferon- gamma; Aim, apoptosis inhibitor of macrophages; TGF-β, transforming growth factor-β; Timp1, tissue inhibitor of metalloprotease 1; Cd68, cluster of differentiation 68; Mcp1, monocyte chemoattractant protein-1; FLT-1, truncated form of the VEGF receptor; ANG2, angiopoietin -2; KDR, tyrosine-protein kinase that acts as a cell-surface receptor for VEGF, TEK, tyrosine kinase, and endothelia; RORγt, RAR-related orphan receptor gamma transcription factor; PNPLA-3, patatin-like phospholipase domain-containing protein 3; Treg/Tr1, regulatory T cell/ type 1 regulatory T cell; T3, triiodothyronine; Th17, T helper 17 cell; SOD, superoxide dismutase; GSH, glutathione; TG, triglyceride; LDLC, low-density lipoprotein cholesterol; FFA, free fatty acid; HOMA-IR, homeostatic model assessment-insulin resistance; ACC-1, acetyl-CoA carboxylase; PPARγ, peroxisome proliferator-activated receptor gamma. DSS, dextran sulfate sodium; HMGB1, high-mobility group box 1; Ki-67, antigen Ki-67; NF-κB, nuclear factor- κB; BMT, bone marrow transplantation; b.w., body weight; STHD, steatohepatitis-inducing high-fat diet; DEN, Diethylnitrosamine; i.p., intraperitoneal injection; i.v., intravenous injection; ITF, Insulin-type fructans; Gro-a, Growth-regulated alpha protein; BrdU, Bromodeoxyuridine; 53BP1, Tumor suppressor p53-binding protein 1; Abx, Antibiotics.

**Table 2 microorganisms-07-00121-t002:** Clinical studies on relation between the microbiome and hepatocellular carcinoma.

	Disease	Comparison	Microbiome Factor	Reference
Human	HCC	HCC patients vs. non-HCC patients	*E. coli*↑	[109]
Human	HCC	non-HBV non-HCV (NBNC)-related HCC vs. HBV-related HCC	*Escherichia-Shigella*↑, *Enterococcus*↑, *Proteus*↑, *Veillonella*↑, *Faecalibacterium*↓, *Ruminococcus*↓, *Ruminoclostridium*↓, *Pseudobutyrivibrio*↓, *Lachnoclostridium*↓, *Phascolarctobacterium*↓	[110]
Human	HCC	486 fecal samples from HCC and cirrhosis patients	*Actinobacteria*↑, *Gemmiger*↑, *Parabacteroides*↓, butyrate-producing genera↓	[78]
Human	HCC	NAFLD-related cirrhosis and HCC, NAFLD-related cirrhosis without HCC, and healthy controls	fecal calprotectin↑, IL 8↑, IL 13↑, chemokines↑	[61]

↑ indicates an increase in condition A relative to condition B, ↓ indicates a decrease in condition A relative to condition B. HBV, hepatitis B virus; HCV, hepatitis C virus; WBC, white blood cells; HCC, hepatocellular carcinoma; NAFLD, nonalcoholic fatty liver disease.

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
