# Peer review of "Role of Gut Microbiota in Hepatocarcinogenesis"

_microorganisms, 2019, doi:10.3390/microorganisms7050121_

Round 1

Reviewer 1 Report

This is a well organized and written manuscript. The references are up to date, the information are presented in a clear way. I have only one small suggestion - as the Authors describe the role of gut microbiota in induction of inflammation and subsequent liver damage, I would suggest the Authors to additionally discuss the potential impact of probiotics on decreasing hepatocyte injury (for instance, lowering serum transaminase activity), ie: Minerva Med. 2018 Dec;109(6):418-428.; J Gastrointestin Liver Dis. 2018 Mar;27(1):41-49. ; Clin Nutr. 2017 Dec;36(6):1530-1539.; Eur Rev Med Pharmacol Sci. 2011 Sep;15(9):1090-5.

Author Response

“Role of gut microbiota in hepatocarcinogenesis”

Point-to-point responses to comments by the Reviewer 1

First of all, we would like to thank the Reviewer1 for his/her comments, which helped us to improve this manuscript.

Specific Comments:

Comment 1: This is a well organized and written manuscript. The references are up to date, the information are presented in a clear way. I have only one small suggestion - as the Authors describe the role of gut microbiota in induction of inflammation and subsequent liver damage, I would suggest the Authors to additionally discuss the potential impact of probiotics on decreasing hepatocyte injury (for instance, lowering serum transaminase activity), ie: Minerva Med. 2018 Dec;109(6):418-428.; J Gastrointestin Liver Dis. 2018 Mar;27(1):41-49. ; Clin Nutr. 2017 Dec;36(6):1530-1539.; Eur Rev Med Pharmacol Sci. 2011 Sep;15(9):1090-5.

Response 1: Thanks for this suggestion and raising this area of concern to discuss the beneficial impact of probiotics in decreasing hepatocyte injury. We discussed the potential impact of probiotics on decreasing hepatocyte injury by reviewer 1. Since our major concerns were microbiota and probiotics intervention on hepatocellular injury, we mainly mentioned serum markers for hepatocyte injury in text body and serum markers in Table 1.

“The potential impact of probiotics on decreasing hepatocyte injury is another factor in the inhibition of HCC in NAFLD [75]. Coadministration of a live multi-strain probiotic mixture with omega-3 fatty acids once daily for 8 weeks to patients with NAFLD reduce liver fat, improve serum lipids, metabolic profile, and reduce chronic systemic inflammatory state [76]. Combination of probiotics Lactobacillus bulgaricus and Streptococcus thermophilus attenuates liver aminotransferases in NAFLD patients [75]. Lowering of proinflammatory marker could restrict the intestinal permeability and translocation of LPS in liver thus alleviating the development of NAFLD and NASH. In another report, the probiotic (Symbiter) reduces liver fat, aminotransferase activity, and the TNF-α and IL-6 levels in NAFLD patients [77]. Probiotics effectively prevents postoperative infections and improves early biochemical parameters of allograft function after liver transplantation [78].”

“Moreover, continuous administration of probiotics prior to liver transplant successively scaled down to 4.8% of 30-day post-transplant infection rate compared to the placebo in post-transplantation in patients [78].”

Reviewer 2 Report

The review of Haripriya Gupta et al. addresses the interesting topic of the relationship between gut microbiota and hepatocellular carcinoma. The review seems to be fairly up to date and well written, only few topics could be addressed further in deep and the general organization could be a little more justified and exposed. 

Some issues are raised:

The role of the compartment of progenitor cells in hepatocarcinogenesis is only indirectly mentioned in relation to original papers that address the role of TLR4 but it would deserve to be more carefully developed.

A concise summary of the review would be useful to insert at the end of the introduction to better illustrate the purpose and importance of the chosen order of the topics covered.

Author Response

“Role of gut microbiota in hepatocarcinogenesis”

Point-to-point responses to comments by the Reviewer 2

First of all, we would like to thank the Reviewer2 for his/her comments, which helped us to improve this manuscript.

To be sure, we checked our manuscript by a professional English editing service of MDPI, in spite of our manuscript did not get suggestion about ‘extensive English Editing’.

Specific Comments:

Comment 1: The review of Haripriya Gupta et al. addresses the interesting topic of the relationship between gut microbiota and hepatocellular carcinoma. The review seems to be fairly up to date and well written, only few topics could be addressed further in deep and the general organization could be a little more justified and exposed.

Some issues are raised: The role of the compartment of progenitor cells in hepatocarcinogenesis is only indirectly mentioned in relation to original papers that address the role of TLR4 but it would deserve to be more carefully developed.

Response 1: We are extremely sorry for writing very brief and compact relationship of TLR4 receptors indirectly with progenitor cells which wasn’t much informative. So, we have added the content in revision and yes we agree with you that it should be mentioned more carefully since progenitor cell in cancer serves as oncogene and here with respect to TLR, after activation of TLR-4, it promoted TLR-4 dependent carcinogeneis.   

“TLR-4 induction and activation are considered to mediate carcinogenesis by a synergistic effect of alcohol and HCV nonstructural protein 5A. A progenitor stem cell marker and TLR-4 downstream gene were also upregulated upon the activation of TLR-4 receptor and aided TLR4-dependent liver carcinogenesis [36].”

Comment 2: A concise summary of the review would be useful to insert at the end of the introduction to better illustrate the purpose and importance of the chosen order of the topics covered.

Response 2: We agree with the reviewer’s comment and apologize for it. We have added a concise summary in last paragraph of the introduction so as to give an outline of our review manuscript that would reflect the subject matter discussed in manuscript. 

The gut microbiota has emerged as a paramount causative event in the progression of hepatic malignancies. Henceforth, being the most pliable organ in human, gut microbiota should be targeted to procure eubiosis. Integrity of microbiome is a desideratum by the therapeutic gut modulation which can be retrieve by therapeutic interventions. Therefore, this review provides an update on the various mechanisms that may show acrimonious communication between the liver and gut microbiota, and how their modulation during pathogenesis contributes to the progression of hepatic diseases to HCC.”
